# Bone metastasis and skeletal-related events in patients with solid cancer: A Korean nationwide health insurance database study

**Soojung Hong**[1]*, **Taemi Youk**[2], **Su Jin Lee**[3], **Kyoung Min Kim**[4], **Claire M. Vajdic**[5]

**1** Division of Oncology-hematology, Department of Internal Medicine, National Health Insurance Service Ilsan Hospital, Goyang, Republic of Korea, **2** Research Institute, National Health Insurance Service Ilsan Hospital, Goyang, Republic of Korea, **3** Department of internal medicine, Seoul Red Cross Hospital, Seoul, Republic of Korea, **4** San Francisco Coordinating Center, California Pacific Medical Center Research Institute, San Francisco, CA, United States of America, **5** Centre for Big Data Research in Health, University of New South Wales, Sydney, NSW, Australia

* suzzy901@nhimc.or.kr, suzzy901@naver.com

**Data Availability Statement:** All files are available from the NHIS-NSC 2002-2013 database.

## Abstract

Bone is one of the most common sites of metastasis from advanced solid tumors. Bone metastasis is a leading cause of pain and increases the risk of skeletal-related events (SREs) in cancer patients. In addition to affecting the quality of life, it also increases the medical costs and mortality risk. We aimed to examine the occurrence of bone metastasis and SREs in Korean cancer patients using a nationwide health database. Using claims data from the National Health Insurance Service-National Sample Cohort (2002–2013), we extracted the data of bone metastasis patients diagnosed with any of the seven major cancers in Korea from January 2002 to December 2010. Selected SREs included pathologic fracture, spinal cord compression, radiation therapy, and palliative bone surgery. We used time-to-event analysis to estimate patient survival after bone metastasis. A total of 21,562 newly diagnosed cancer patients were identified; bone metastases developed in 1,849 patients (breast cancer, 18.8%; prostate cancer, 17.5%; lung cancer, 13.7%). The median time from primary cancer diagnosis to bone metastasis was 18.9 months. The cumulative incidence of SREs was 45.1% in all bone metastasis patients. The most common cancer type was lung cancer (53.4%), followed by liver (50.9%), prostate (45.9%), breast (43.6%), and colorectal (40.2%) cancers. Almost all SREs developed 1 month after bone metastasis, except in patients with breast and prostate cancers (median: 5.9 months in breast cancer and 4.7 months in prostate cancer). Survival duration after the development of bone metastasis was < 6 months in stomach, liver, colorectal, and lung cancer patients. Breast and prostate cancer patients survived for > 1 year after the occurrence of SREs. This study reveals the epidemiology of bone metastasis and SREs in Korean cancer patients, and the findings can be used to assess the actual bone health status of cancer patients.

**Funding:** S.H. supported by the National Health Insurance Service Ilsan Hospital Grant (NHIS-2016-20-020).

**Competing interests:** The authors have declared that no competing interests exist.

## Introduction

Bone is one of the most common sites of metastasis from advanced solid cancers, and bone metastases occur in 65–80% of patients with advanced prostate or breast cancer, 40–50% patients with lung cancer, and in <10% of those with gastrointestinal cancer [1–3]. The bone is a dynamic organ that undergoes constant remodeling with simultaneous bone formation and resorption by osteoblasts and osteoclasts, respectively. In a normal healthy bone, the interaction between osteoblasts and osteoclasts is well balanced. However, once cancer cells invade the bone tissue, bone remodeling balance is disrupted, leading to destruction of the skeleton [4]. In addition, cancer cells that have metastasized to the bone secrete cytokines and growth factors causing a change in the microenvironment. Bone can also serve as a reservoir for dormant cancer cells from other organs, which can lead to full-blown metastases after a long period of dormancy [3].

Bone metastasis is a leading cause of pain in cancer patients. Bone pain due to metastasis is mainly caused by destruction of the bone structure, leading to periosteal irritation and nerve damage [5]. Moreover, it increases the risk of complications referred to as skeletal-related events (SREs), including pathologic fracture, spinal cord compression, palliative radiation to the bone, and palliative bone surgery [6]. Bone metastases and SREs greatly affect the quality of life of patients, in addition to increasing the medical costs [7], and mortality risk [1, 8].

Various studies have reported the development of bone metastases from breast, prostate, and lung cancers. However, with an increasing number of cancer survivors, bone metastasis has become a critical issue, and the incidence of bone metastasis in other type of cancer should be investigated as well. Moreover, there exists a cross-country variation in the pattern and occurrence of bone metastasis from each type of primary cancer. There are a limited number of population-based real-world studies reporting on the occurrence of bone metastasis in patients with solid cancers [9–11]. This study aimed to investigate the occurrence and pattern of bone metastases and SREs in selected patients with solid cancers using the Korean nationwide health insurance cohort database.

## Materials and methods

### Data source

The Korean National Health Insurance Service (KNHIS) enrolls more than 97% of the entire Korean population. The KNHIS established a national sample cohort, the National Health Insurance Service-National Sample Cohort (NHIS-NSC), for research purposes that was randomly selected and comprised 2.2% of the total eligible population [12]. We used the NHIS-NSC claims data from 2002 to 2012. This database provides detailed information on demographics and healthcare utilization, including diagnostic codes (International Classification of Diseases 10th revision, ICD-10), procedure codes, and prescriptions.

### Study population

We identified all newly diagnosed patients with primary stomach, colorectal, liver, lung, breast, prostate, and other genitourinary (GU) cancers, which were considered as major types of cancers in Korea [13] from January 2003 to December 2010 (ICD-10 codes: C16, C18–20, C22, C34, C50, C61, C51–57, and C60–68). We used a 1-year washout period to exclude cancer patients who had been diagnosed in the past. Other GU cancers included malignant neoplasms of the female genital organs, male genital organs, and the urinary tract. We excluded participants who were aged less than 20 years and those with multiple primary cancers. Patients were followed up for at least 2 years to assess for the development of bone metastases and SREs.

Patients diagnosed with bone metastases were identified using a diagnostic code (ICD-10 code for bone metastasis: C795). SREs were identified using the diagnostic codes for pathologic fracture (ICD-10 code: S22, S32, S42.2–42.4, S52, S62, S72, S82.1–82.4, T08, T12, and M80.0) and spinal cord compression (ICD-10 code: G95.2, G55.0, and G55.8), and the procedure codes for bone radiation (HD051–HD061, HD080–HD089, HD110–HD112, HD211, HD212, HD121, and HZ271) and bone surgery (N0304-N0309, N0451-N0453, N0466-N0469, N0471-N0475, N0590, N0601-N0617, N0630, N0641-N0645, N0981-N0986, N0991-N0995, N1466, N1469, N1497-N1499, N2461-N2470, N2491-N2499, S4694-S4696, S4704-S4709, S6691-S6696).

## Statistical analysis

Data on the occurrence of bone metastasis in each type of cancer and basic demographic characteristics of the patients were obtained. The median overall survival was calculated from the date of primary cancer diagnosis to the date of death. We also calculated the median survival after bone metastasis, which was defined as the period from the diagnosis of bone metastasis to death. Time-to-event analysis was performed to estimate the time interval from the development of bone metastasis to the occurrence of SREs, or to estimate the cumulative incidence of SREs due to bone metastasis. Survival analysis was performed using the Kaplan–Meier method. All analyses were performed using SAS version 9.4 (SAS Institute Inc., Cary, NC, USA).

## Statement of ethics

This study was approved by the Institutional Review Board of the National Health Insurance Service Ilsan Hospital (NHIMC 2016-03-007), and the requirement for written informed consent was waived.

## Results

We identified 21,562 patients with one of the seven types of cancers diagnosed between 2003 and 2010 who were followed up until the end of 2012 (Table 1). Among them, 1,849 patients had been newly diagnosed with bone metastases at or after the diagnosis of the primary cancer, accounting for 8.6% of the entire cohort. Bone metastases were commonly observed in breast cancer (18.8%), prostate cancer (17.5%), and lung cancer (13.7%) patients

Table 1. Selection of patients with cancer and bone metastasis.

| | Criteria | Stomach | Colorectal | Liver | Lung | Breast | Prostate | Other GU | Total |
|---|---|---|---|---|---|---|---|---|---|
| Step 1 NHIS-NSC 2002–2012 cancer diagnosis (n) | | 8,754 | 7,468 | 5,527 | 6,383 | 4,380 | 2,411 | 6,385 | 41,308 |
| Step 2 No evidence of any other primary cancer (n) | | 7,706 | 6,353 | 4,793 | 5,555 | 4,063 | 1,940 | 5,590 | 36,000 |
| Step 3 New diagnosis from 2003to 2010 and age >20 years (n) | | 4,653 | 3,860 | 3,024 | 3,489 | 2,221 | 1,109 | 3,206 | 21,562 |
| Step 4 New diagnosis of bone metastasis (n, %) | | 190 (4.1%) | 169 (4.4%) | 173 (5.7%) | 479 (13.7%) | 417 (18.8%) | 194 (17.5%) | 227 (7.1%) | 1,849 (8.6%) |

GU, genitourinary; NHIS-NSC, National Health Insurance Service-National Sample Cohort

Table 2 shows the characteristics of patients with bone metastasis according to the primary cancer type. For the majority of the cancers, except other GU cancers, bone metastases were more common in men than in women. Among patients with stomach, colorectal, and lung cancer, bone metastases frequently occurred in patients who were in their sixties. In prostate cancer, the incidence of bone metastases increased with age, whereas in breast cancer, it decreased with age. Bone metastases showed an increasing trend in rural areas compared to that in urban areas. Household income had no impact on the occurrence of bone metastasis. The diagnosis of bone metastasis was made together with the initial cancer diagnosis in 46.3% of the patients. The incidence of bone metastasis was the highest in lung cancer patients (64.3%), followed by those with breast cancer (47.7%) and prostate cancer (47.4%). For metachronous bone metastasis, the average time between primary cancer diagnosis and bone metastasis was 18.9 months (standard deviation: 21.6 months). The shortest interval between the diagnoses of primary cancer and bone metastasis was seen in lung cancer patients at 9 months, followed by breast cancer patients at 14.9 months, and prostate cancer patients at 17.4 months

Among the 1,849 patients with bone metastases, 833 (45.1%) experienced SREs (Table 3). SREs most commonly occurred in patients with lung cancer (53.4%), followed by those with

**Table 2. Characteristics of patients with bone metastasis according to the primary cancer type.**

| Characteristics (no., %) | | Stomach (N = 190) | Colorectal (N = 169) | Liver (N = 173) | Lung (N = 479) | Breast (N = 417) | Prostate (N = 194) | Other GU (N = 227) | Total (N = 1,849) | p-value |
|---|---|---|---|---|---|---|---|---|---|---|
| Sex | | | | | | | | | | |
| | Male | 116 (61.1%) | 91 (53.8%) | 128 (74.0%) | 354 (73.9%) | 1 (0.2%) | 194 (100%) | 97 (42.7%) | 981 | <0.01 |
| | Female | 74 (38.9%) | 78 (46.2%) | 45 (26.0%) | 125 (26.1%) | 416 (99.8%) | 0 | 130 (57.3%) | 868 | |
| Age group (years) | | | | | | | | | | |
| | 20–49 | 45 (23.7%) | 32 (18.9%) | 43 (24.9%) | 47 (9.8%) | 223 (53.5%) | 6 (3.1%) | 57 (25.1%) | 453 | <0.01 |
| | 50–59 | 45 (23.7%) | 34 (20.1%) | 56 (32.4%) | 100 (20.9%) | 124 (29.7%) | 21 (10.8%) | 57 (25.1%) | 437 | |
| | 60–69 | 58 (30.5%) | 56 (33.1%) | 41 (23.7%) | 176 (36.7%) | 48 (11.5%) | 73 (37.6%) | 64 (28.2%) | 516 | |
| | ≥70 | 42 (22.1%) | 47 (27.8%) | 33 (19.1%) | 156 (32.6%) | 22 (5.3%) | 94 (48.5%) | 49 (21.6%) | 443 | |
| Region | | | | | | | | | | |
| | Urban | 79 (41.6%) | 75 (44.4%) | 64 (37.0%) | 202 (42.2%) | 223 (53.5%) | 87 (44.8%) | 106 (46.7%) | 836 | <0.01 |
| | Rural | 111 (58.4%) | 94 (55.6%) | 109 (63.0%) | 277 (57.8%) | 194 (46.5%) | 107 (55.2%) | 121 (53.3%) | 1,013 | |
| Income | | | | | | | | | | |
| | 1%–40% | 62 (32.6%) | 53 (31.4%) | 46 (26.6%) | 149 (31.1%) | 116 (27.8%) | 49 (25.3%) | 62 (27.3%) | 537 | 0.18 |
| | 41%–80% | 70 (36.8%) | 62 (36.7%) | 65 (37.6%) | 189 (39.5%) | 177 (42.4%) | 64 (33.0%) | 95 (41.9%) | 722 | |
| | 80%–100% | 58 (30.5%) | 54 (32.0%) | 62 (35.8%) | 141 (29.4%) | 124 (29.7%) | 81 (41.8%) | 70 (30.8%) | 590 | |
| Bone metastasis | | | | | | | | | | |
| | At the initial cancer diagnosis | 59 (33.1%) | 43 (25.4%) | 79 (45.7%) | 308 (64.3%) | 199 (47.7%) | 92 (47.4%) | 77 (33.9%) | 857 (46.3%) | |
| | Time to bone metastasis (Mean, SD) (month) | 23.4 (22.6) | 28.9 (25.5) | 16.3 (21.2) | 9.0 (15.2) | 14.9 (20.6) | 17.4 (22.0) | 22.6 (23.8) | | |

GU, genitourinary; SD, standard deviation

**Table 3.  Overall occurrence of skeletal-related events (number of patients).**

| Characteristics (No., %) | Stomach | Colorectal | Liver | Lung | Breast | Prostate | Other GU | Total |
|---|---|---|---|---|---|---|---|---|
| | (N = 190) | (N = 169) | (N = 173) | (N = 479) | (N = 417) | (N = 194) | (N = 227) | (N = 1,849) |
| Total patients with SREs | 72 (37.9%) | 68 (40.2%) | 88 (50.9%) | 256 (53.4%) | 182 (43.6%) | 89 (45.9%) | 78 (34.4%) | 833 (45.1%) |
| Fracture | 27 (14.2%) | 20 (11.8%) | 13 (7.5%) | 33 (6.9%) | 50 (12.0%) | 35 (18.0%) | 23 (10.1%) | 201 (10.9%) |
| Cord compression | 11 (5.8%) | 11 (6.5%) | 6 (3.5%) | 11 (2.3%) | 8 (1.9%) | 6 (3.1%) | 10 (4.4%) | 63 (3.4%) |
| Radiation | 34 (17.9%) | 31 (18.3%) | 65 (37.6%) | 215 (44.9%) | 142 (34.1%) | 55 (28.4%) | 45 (19.8%) | 587 (31.7%) |
| Surgery | 17 (8.9%) | 18 (10.7%) | 21 (12.1%) | 46 (9.6%) | 21 (5.0%) | 18 (9.3%) | 13 (5.7%) | 154 (8.3%) |
| Median time from BM to SREs (months) | 0.5 | 0.4 | 0.0 | 0.4 | 5.9 | 4.7 | 0.1 | |

GU, genitourinary; BM, bone metastasis; SREs, skeletal-related events

liver cancer (50.9%), prostate cancer (45.9%), breast cancer (43.6%), and colorectal cancer (40.2%). The most common SRE was radiation therapy to the bone, which was administered to 31.7% of patients who developed bone metastasis and to 70.5% of patients with SREs. Patients with lung cancer (44.9%) and liver cancer (37.6%) more frequently received radiation therapy for bone metastasis. Fractures occurred in 10.9% of all patients with bone metastases and were the most frequent in prostate cancer (18.0%). A total of 8.3% patients underwent bone surgery and 3.4% experienced spinal cord compression. Fig 1. shows the total number of SREs according to the cancer type. The majority of SREs occurred within 1

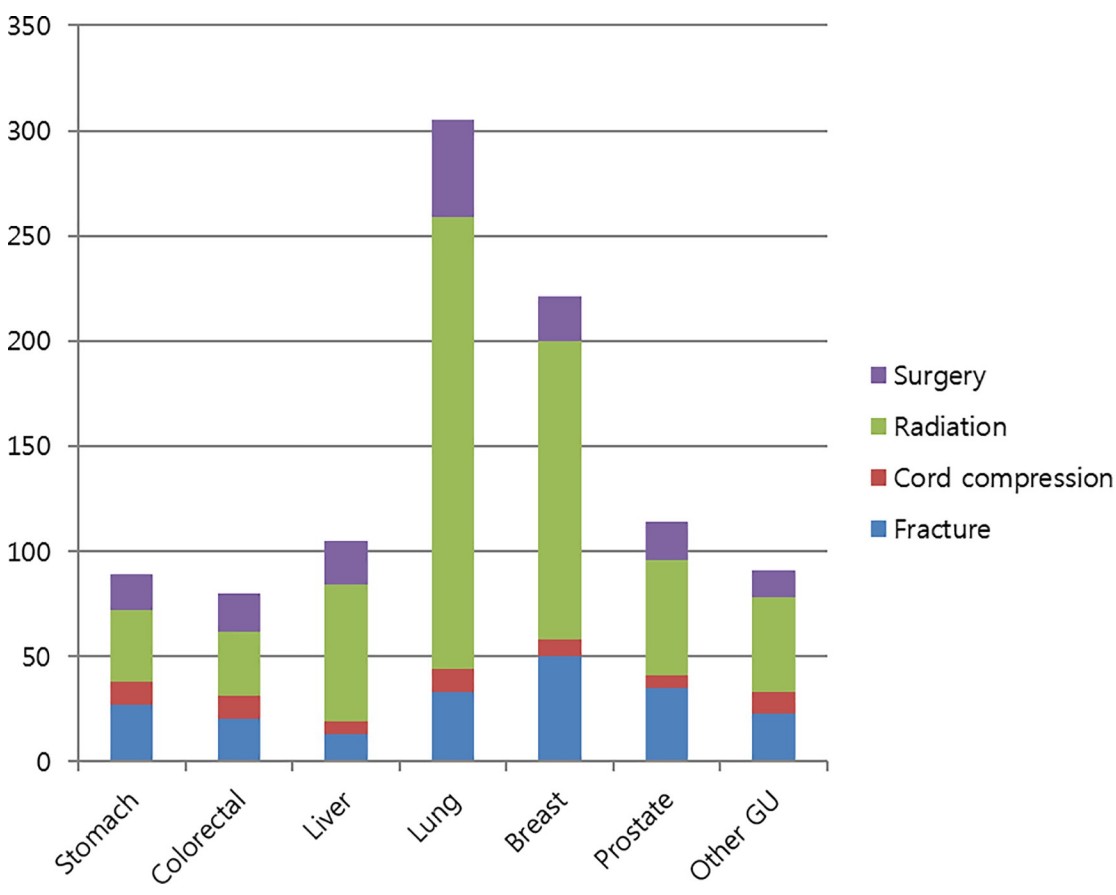

**Fig 1.  Number of skeletal-related events.**

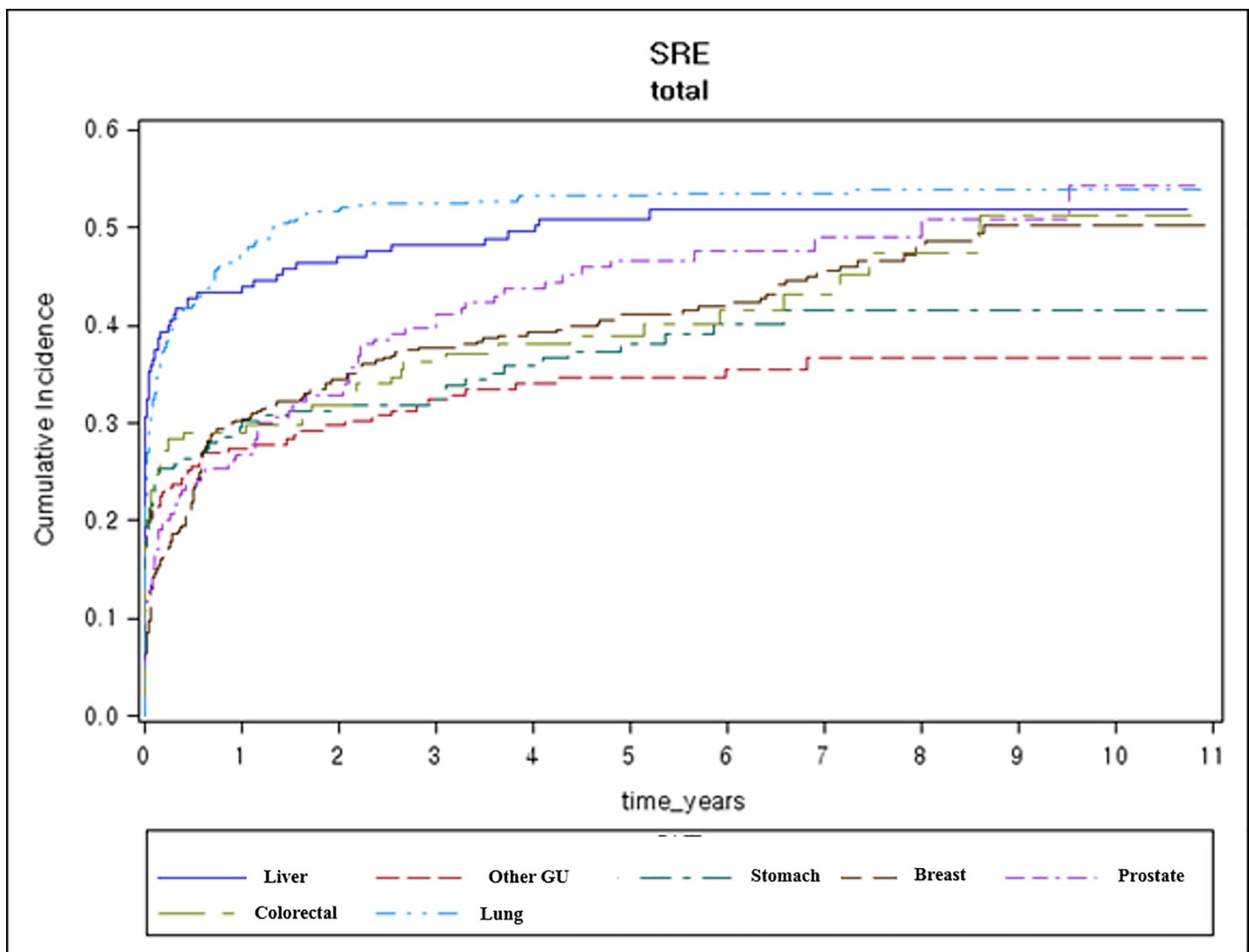

**Fig 2. Cumulative risk of skeletal-related events (%).**

month of bone metastasis, except in breast (5.9 months) and prostate (4.7 months) cancers. SREs were significantly more frequent in breast cancer patients when the bone metastasis was diagnosed simultaneously with the primary cancer (p < 0.001). We estimated the cumulative incidence of SREs (Fig 2). The estimated cumulative incidence of SREs is summarized in S1 Table.

The overall survival of patients varied with the primary cancer type, with OS being the shortest in lung cancer (10 months) and the longest in breast cancer (39 months). The median survival after bone metastasis was 3.5 months in stomach cancer patients, 4 months in liver cancer patients, 5 months in lung and colorectal cancer patients, 9 months in other GU cancer patients, 16 months in prostate cancer patients, and 19 months in breast cancer patients (Fig 3). Fig 4 and Fig 5 present Kaplan-Meier overall survival curve for whole cancer patients and survival after bone metastasis, respectively.

**Fig 3. Overall survival and survival after the diagnosis of bone metastasis.**

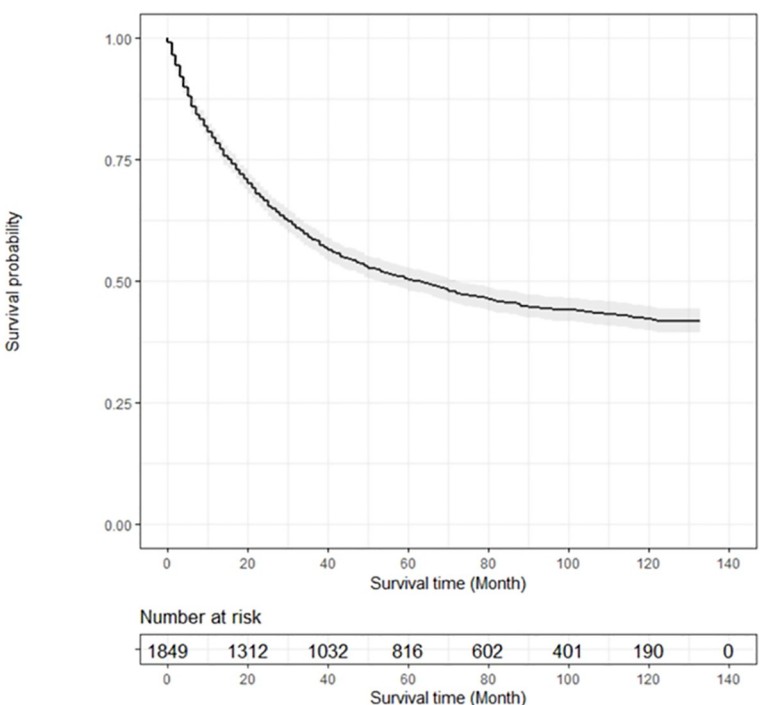

**Fig 4. Kaplan-Meier survival curve for overall survival.**

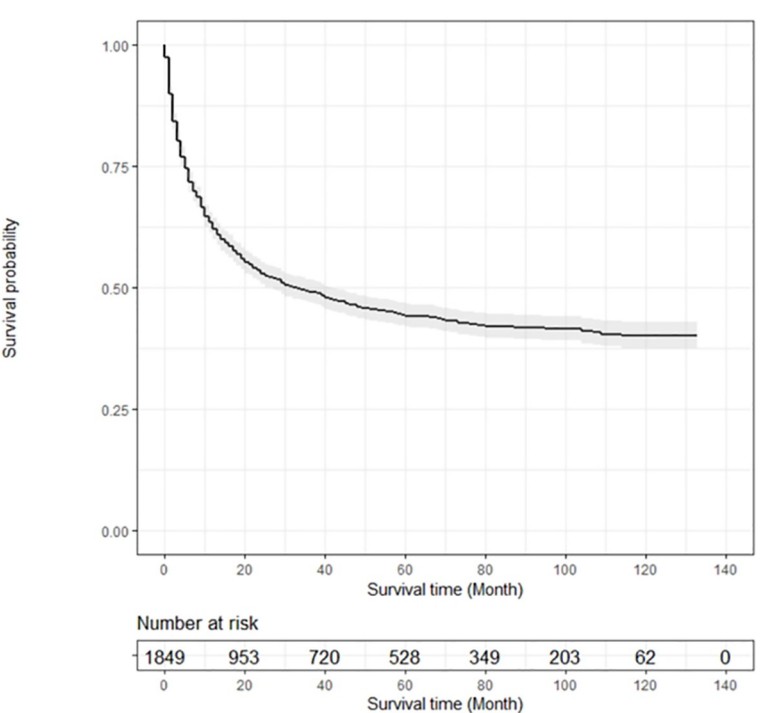

**Fig 5. Kaplan-Meier survival curve for overall survival after bone metastasis.**

## Discussion

Our study included 1,849 patients with bone metastases arising from seven major types of cancer who were enrolled from the Korean National Sample Cohort database. We found that 8.6% of the newly diagnosed cancer patients had bone metastases. The overall occurrence of bone metastasis from solid cancers was similar to that reported from other countries. In 2004, Schulman et al. reported that 5.3% of cancer patients in the United States (US) had metastatic bone disease [7]. A recent US study identified patients with solid cancer from 2004 to 2013 and reported that 6.9% of the entire study population had bone metastases at diagnosis and during follow-up [11]. A Danish population-based cohort study registered patients diagnosed with cancer between 1994 and 2010 and showed that 7.5% of patients with prostate, breast, and lung cancer, which were the three most common primary cancers out of 10 cancer types, had metastatic bone disease [10]. Similarly, a retrospective cohort study conducted in Thailand from 2006 to 2015 demonstrated that 7.7% of the patients diagnosed with the 10 most common cancer types had bone metastases [9]. The incidence of bone metastasis according to the cancer type differed between countries. However, most previous studies, including our study, showed that the top three leading cancers associated with the development of bone metastasis were prostate, breast, and lung cancers [9–11]. In comparison, less than 5% of stomach and colorectal cancer patients developed bone metastasis, which is consistent with the historical data (3%–5%) [2].

We found that the mean time from primary cancer diagnosis to bone metastasis was 18.9 months, and this is consistent with the findings of a US study, which reported that the incidence of bone metastasis in patients with solid cancer mostly increased in the first 2 years [11]. According to our data, lung cancer patients demonstrated the shortest interval between the

diagnoses of primary cancer and bone metastasis and also the highest proportion (64.3%) of concomitant diagnoses for primary tumor and bone metastasis. In a US study, 64.4% of lung cancer patients were diagnosed with bone metastases at the time of the initial cancer diagnosis [14]. This may be due to the rapid and aggressive metastatic process in lung cancer, which can be attributed to the lung cancer cells swiftly obtaining the ability to infiltrate and colonize other organs [15]. However, in other cancers, especially in colorectal cancer, there was a longer interval to overt bone metastasis (mean: 28.9 months), and the incidence of simultaneous diagnosis of the primary tumor and bone metastasis was only 25.4%. These findings suggest that there are different mechanisms underlying the development of bone metastases in different cancer types, and these mechanisms depend on the patient's oncogenic background as well as the microenvironment in each type of cancer [15].

Among all patients with bone metastasis, 45.1% developed SREs. With regard to lung and liver cancers, SREs occurred in more than 50% patients, whereas in breast, prostate, and colorectal cancer patients, about 40% or more developed SREs. In previous breast cancer trials, a total of 64% of patients who did not use bone-modifying agents (BMAs) experienced SREs, and 43–45% of patients who used BMA developed SREs [16, 17]. In case of prostate cancer, approximately 44–49% of patients in the placebo group developed SREs, while 33–41% of the patients in the BMA group developed SREs [18–20]. Regarding lung cancer and other solid cancers, previous randomized trials have revealed that 44–46% of patients in the placebo group and 35–39% of patients in the BMA group developed SREs [21, 22]. Compared with those of placebo-controlled clinical trials, our real-world data showed a higher incidence of SREs in lung cancer patients and a similar or lower incidence of SREs in breast and prostate cancer patients. In addition, compared with that in recent clinical trials on novel BMAs (such as zoledronate or denosumab), the occurrence of SREs in our routine practice was much higher. A pooled analysis of phase 3 trials showed overall SRE incidence rate of 33.5%, 38.2%, and 28.5% in patients with breast, prostate, and other solid cancers, respectively [23]. This difference might be because of the insurance coverage in Korea as well as the differences in clinical practice in the real world. However, there is a lack of data on stomach cancer, which has the highest incidence in Korea. A previous Italian multicenter study reported an overall SRE incidence of 31% in stomach cancer patients, and radiation therapy comprised 47.1% of the SREs [24]. The Memorial Sloan Kettering Cancer Center (MSKCC) evaluated 459 patients with liver cancer. Among them, 32.9% had bone metastasis, and the overall occurrence of SREs in patients with bone metastasis was 56.3% [25]. These findings are similar to our results in patients with stomach or liver cancer. The most common SRE was radiation therapy (31.7%), which has been frequently reported in other Korean studies [26, 27], but is less frequently reported in other countries [2, 28]. The second commonest SRE was pathologic fracture, accounting for 10.9% of the cases, and more than half of these SREs were fractures of the spine or pelvis.

The estimated 3-year cumulative incidence of SREs was 45–55% in all cancer types. The 1-year cumulative SRE incidence rates in lung cancer (47.8%), breast cancer (32.1%), and prostate cancer (29.9%) patients were relatively low compared to other countries. According to data from a US study, the SRE incidence rate in both lung cancer and breast cancer patients was 45.4%, while that in prostate cancer patients was 30.4% [14]. A Danish population study reported an SREs incidence rate of 55.0% in lung cancer [8], 46.1% in prostate cancer [29], and 38.5% in breast cancer [30]. A longitudinal view of the cumulative incidence of SREs (Fig 2) showed the most significant change over time in case of prostate cancer. Therefore, patients with prostate cancer should be carefully followed up for SREs because they show longer survival than those with other cancer types of cancers.

The median time from the diagnosis of bone metastasis to SREs was within 1 month, except in breast and prostate cancer. An Italian study reported that the median time from bone metastasis to the occurrence of SREs was 2 months in colorectal cancer, and the survival duration after SREs was estimated to be 4.5 months [28]. The MSKCC group reported that in liver cancer patients, the median time from bone metastasis to the development of SREs was 0.8 months. Meanwhile, an Italian study reported that in stomach cancer patients, the median time from bone metastasis to the development of SREs was 2 months [24, 25]. These findings are remarkably similar to our results. Our study patients only survived 3–5 months after the occurrence of SREs. Consequently, following the diagnosis of bone metastasis, physicians should closely follow-up the development of SREs and manage them in a timely manner.

We estimated the median survival after the diagnosis of bone metastasis along with the overall survival for each cancer type. The overall survival depends predominantly on the prognosis of the primary cancer; it was longer in breast and prostate cancer patients and shorter in lung cancer patients. Regarding median survival after the diagnosis of bone metastasis, it was the shortest in stomach cancer patients (3.5 months), followed by liver cancer (four months), lung cancer, and colorectal cancer (five months) patients. Generally, the majority of these cancers had a poor prognosis. However, in colorectal cancer, the prognosis was unusual. The median overall survival in colorectal cancer was 23 months, which was longer than that in other cancer, but the median survival after bone metastasis was poor. An Italian study also reported that the period from bone metastasis to death was only 7 months in colorectal cancer patients [28]. In contrast, in breast and prostate cancers, bone metastases developed in the middle of the disease course. Since, patients with these cancers tended to live for more than 1 year, and they had more chances of experiencing SREs [27]. Accordingly, a multidisciplinary team approach should be used for the management of breast and prostate cancer patients to provide supportive care for bone metastases along with potential aggressive treatment for SREs such as fractures and spinal cord compression.

This study has several limitations. Although 2% of the national sample cohort database was utilized in this study, the patients were selected based on the diagnostic code, which may be less accurate than the data from chart reviews or cross-sectional studies; therefore, the number of patients could have been underestimated. Consequently, caution is required when generalizing our data to the whole Korean population. For the definition of radiation therapy, we could not distinguish the reasons for radiation therapy because we used the claims data. Radiation therapy is usually administered when bone pain is present, but sometimes, it is also used for asymptomatic patients to reduce the risk of SREs. Lastly, we could not obtain specific cancer data such as those on cancer stage and subtype from the claims data.

However, the results of our study are significant because they show the current incidence and trends of bone metastasis and SREs in Korean patients with solid cancers and provide physicians valuable insight into this patient population. An additional strength of our study is that it presents the epidemiology and natural history of bone metastasis in stomach and liver cancer, both of which are prevalent in Asia. With the development of cancer therapy and an increase in the rates of early diagnosis, the number of long-term cancer survivors continues to rise, and maintaining the well-being of these patients is a major issue in cancer care. Therefore, physicians should consider active management of bone metastases to control related symptoms and to prevent SREs. The choice of treatment should be based on the course of the disease and performance status of the patient.

## Conclusion

In summary, this study estimated the occurrence of bone metastasis and subsequent SREs in selected Korean patients diagnosed with various types of solid cancers. We also examined the

timing of bone metastasis and SREs during the course of the disease. We found that the development and timing of bone metastasis and SREs differed according to the type of primary cancer. Metastatic bone disease arising from solid cancer is incurable. Therefore, maintaining the quality of life of the patients and providing full supportive care are of utmost importance. Our data can be used as a basis for planning the management of patients with bone metastases from solid cancers.

## Supporting information

**S1 Table. Cumulative risk of SREs (%).**
(DOCX)

## Acknowledgments

We would like to thank Editage (www.editage.co.kr) for English language editing.

## Author Contributions

**Conceptualization:** Soojung Hong, Su Jin Lee, Kyoung Min Kim.

**Data curation:** Taemi Youk, Su Jin Lee, Kyoung Min Kim.

**Formal analysis:** Soojung Hong, Taemi Youk.

**Funding acquisition:** Soojung Hong.

**Investigation:** Soojung Hong, Su Jin Lee, Kyoung Min Kim.

**Methodology:** Soojung Hong, Taemi Youk.

**Project administration:** Soojung Hong.

**Resources:** Taemi Youk.

**Software:** Taemi Youk.

**Supervision:** Claire M. Vajdic.

**Visualization:** Soojung Hong.

**Writing – original draft:** Soojung Hong.

**Writing – review & editing:** Soojung Hong, Su Jin Lee, Kyoung Min Kim, Claire M. Vajdic.

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
