## [Decision Letter · Decision Letter 0]

4 May 2020

PONE-D-20-03800

Bone metastasis and skeletal-related events in patients with solid cancer: a Korean nationwide health insurance database study

PLOS ONE

Dear Dr Hong,

Thank you for submitting your manuscript to PLOS ONE. After careful consideration, we feel that it has merit but does not fully meet PLOS ONE’s publication criteria as it currently stands. Therefore, we invite you to submit a revised version of the manuscript that addresses the points raised during the review process.

ACADEMIC EDITOR: The manuscript requires a revision according to the recommendations from the reviewers. One reviewer mentioned that the article is not easy-to-read. To ensure the quality of English and the readability of the article, please provide proof of English editing service by native English speakers when resubmitting the revision. 

We would appreciate receiving your revised manuscript by Jun 18 2020 11:59PM. To enhance the reproducibility of your results, we recommend that if applicable you deposit your laboratory protocols in protocols.io, where a protocol can be assigned its own identifier (DOI) such that it can be cited independently in the future. For instructions see: http://journals.plos.org/plosone/s/submission-guidelines#loc-laboratory-protocols

We look forward to receiving your revised manuscript.

Kind regards,

Jason Chia-Hsun Hsieh, M.D. Ph.D

Academic Editor

PLOS ONE

Additional Editor Comments (if provided):

The manuscript requires a revision according to the recommendations from the reviewers. One reviewer mentioned that the article is not easy-to-read. To ensure the quality of English and the readability of the article, please provide proof of English editing service by native English speakers when resubmitting the revision.

2. In your ethics statement in the manuscript and in the online submission form, please provide additional information about the patient records used in your retrospective study. Specifically, please ensure that you have discussed whether all data were fully anonymized before you accessed them.

3. To comply with PLOS ONE submission guidelines, in your Methods section, please provide additional information regarding your statistical analyses. For more information on PLOS ONE's expectations for statistical reporting, please see https://journals.plos.org/plosone/s/submission-guidelines.#loc-statistical-reporting.

Reviewers' comments:

Reviewer's Responses to Questions

**Comments to the Author**

1. Is the manuscript technically sound, and do the data support the conclusions?

Reviewer #1: Yes

Reviewer #2: Yes

Reviewer #3: Yes

Reviewer #4: Yes

Reviewer #5: Partly

2. Has the statistical analysis been performed appropriately and rigorously? 

Reviewer #1: Yes

Reviewer #2: Yes

Reviewer #3: Yes

Reviewer #4: Yes

Reviewer #5: No

3. Have the authors made all data underlying the findings in their manuscript fully available?

Reviewer #1: Yes

Reviewer #2: Yes

Reviewer #3: Yes

Reviewer #4: Yes

Reviewer #5: Yes

4. Is the manuscript presented in an intelligible fashion and written in standard English?

Reviewer #1: Yes

Reviewer #2: Yes

Reviewer #3: Yes

Reviewer #4: Yes

Reviewer #5: No

5. Review Comments to the Author

Reviewer #1: The enclosed manuscript by Hong et al. is an analysis of bone metastases and skeletal related events in Korea using a national database. The authors find an incidence of bone metastases and SREs in an Asian population similar to what has been published from other parts of then world. The authors look at survival after bone mets and SREs and provide useful information on the natural history of bone metastases. The incidence of bone metastases in stomach and liver cancer is useful as these diseases occur frequently in this population.

I believe this article adds to the worldwide information on the scope of the problem of bone metastases.

I suggest acceptance of this manuscript with no revision needed

Reviewer #2: This study is a good dataset may contribute to the better identification of bone health in patients with solid cancers. However, I have some concerns on the methodology.

Specific Comments

1. Which ICD-code of bone surgery SRE in this study ?

2. The literature showed that once a patient experiences an SRE, the risk of subsequent SRE is increase. I wonder if this study had the data on the cumulative number of SREs in each patient ?

3. Pathologic fracture is strongly associated with an increased risk of death. How about the differences of overal survival between BM patients with fracture SRE and those without fracture SRE, and those without SRE.

4. In the discussion part, please correct "1,845 bone metastasis patients" into "1,849".

Reviewer #3: The definition of SRE (Skeletal-related events) about radiation is slightly different for each paper. (severe bone pain requiring radiation vs. just radiation to bone)

In some cases, radiation therapy is administered for asymptomatic bone metastases in patients to reduce the risk of skeletal-related events. Therefore, it must be noted and the authors should concern and describe about this.

Reviewer #4: This is well written and organized paper. The authors addressed the bone metastasis and SRE in patients with diverse solid cancers in Korea.

Minor points

1. In material and methods section. I can’t easily understand the following sentences. Please explain. “We excluded paticipants who were aged less than 20 years~ at least 2 years to assess for bone metastases and SREs” What does washout mean?

2. Please add Kaplan-Meier survival curves for OS, and survival after bone metastasis if possible. (You can add just 2 figures, one for OS including all cancer types, and the other for survival after bone metastasis)

Reviewer #5: This study showed the epidemiology of bone metastases and skeletal-related events (SREs) in Korea using claims data from the National Health Insurance Service National Sample Cohort. However, the study presented only numerical value. Statistic data was not connected to QOL or medical cost, although authors mentioned in discussion and conclusion. I can’t discover a new and unique point from this study. Therefore, unfortunately, I think that this manuscript is not candidate for publication in PLOS ONE.

Major point

1. This study focused on 7 major cancer types (stomach, colorectal, liver, lung, breast, prostate, and other genitourinary cancers). The reference about Korea cancer incidence should be added.

2. Why did authors use the only C795 for bone metastases? The reason should be documented in discussion.

3. Authors defined pathologic fracture, spinal cord compression, bone radiation, and bone surgery as SREs. Why did authors exclude hypercalcemia from SREs? And what was the procedure cord for bone surgery? How did authors handle overlapping data from the different cords? I don’t understand this sentence “In addition, patients diagnosed in 2002 were excluded for washout and were followed up for at least 2 years to assess for bone metastases and SREs”.

4. In result, authors commented that bone mets increased with age and there was no difference in region or household. Statistical p value should be added. If authors did not use statistical analysis, the content should be corrected.

5. Discussion is ambiguous and extensive. This cloud the issue. Please clarify authors’ suggestion and present specified contents.

6. The manuscript is not easy to read and the English used really should be checked for grammar and spelling.

6. PLOS authors have the option to publish the peer review history of their article (what does this mean?). If published, this will include your full peer review and any attached files.

Reviewer #1: No

Reviewer #2: Yes: Lan T. Ho-Pham

Reviewer #3: Yes: IK JAE LEE

Reviewer #4: Yes: Jun Eul Hwang

Reviewer #5: No

---

## [Author Response · Author response to Decision Letter 0]

27 May 2020

1. Which ICD-code of bone surgery SRE in this study?

Thank you for comment. I added procedure codes for bone surgery in study population.

2. The literature showed that once a patient experiences an SRE, the risk of subsequent SRE is increase. I wonder if this study had the data on the cumulative number of SREs in each patient?

Thank you for your comment. I agree with your opinion, however we focused only on the first event of each SRE in this paper. We do not have information about cumulative number of SREs.

3. Pathologic fracture is strongly associated with an increased risk of death. How about the differences of overall survival between BM patients with fracture SRE and those without fracture SRE, and those without SRE.

Thank you for your comment. 

We tried to compare the overall survival according to each SRE, but the proportion of death in each group was small and we could not find any meaningful results.

4. In the discussion part, please correct "1,845 bone metastasis patients" into "1,849".

Thank you for your comment. I corrected it.

Reviewer #3: 

The definition of SRE (Skeletal-related events) about radiation is slightly different for each paper. (severe bone pain requiring radiation vs. just radiation to bone)

In some cases, radiation therapy is administered for asymptomatic bone metastases in patients to reduce the risk of skeletal-related events. Therefore, it must be noted and the authors should concern and describe about this.

Thank you for your comment.

We added your comment for the weaknesses of the discussion. Two reasons for radiation cannot be distinguished from claims data.

“In the definition of radiation therapy, we could not distinguish the reason for radiation therapy because we used the claim data. The radiation therapy is usually performed when bone pain is present, but in some cases, it is also used for asymptomatic patients to reduce the risk of SRE. “

Reviewer #4: 

1. In material and methods section. I can’t easily understand the following sentences. Please explain. “We excluded paticipants who were aged less than 20 years~ at least 2 years to assess for bone metastases and SREs” What does washout mean?

Our database is from 2002. In the first year, usually we use wash out period for cancer diagnosis because previous prevalent cases may confound incidence. We revised the sentence;

“We used a 1-year washout period to exclude cancer patients who had been diagnosed in the past.”

2. Please add Kaplan-Meier survival curves for OS, and survival after bone metastasis if possible. (You can add just 2 figures, one for OS including all cancer types, and the other for survival after bone metastasis)

Thank you for your comment.

We added two figures.

Reviewer #5: 

1. This study focused on 7 major cancer types (stomach, colorectal, liver, lung, breast, prostate, and other genitourinary cancers). The reference about Korea cancer incidence should be added.

We inserted reference about cancer statistics in Korea.

2. Why did authors use the only C795 for bone metastases? The reason should be documented in discussion.

C795 is ICD-10 code for bone metastasis. We added an explanation for the code.

3. Authors defined pathologic fracture, spinal cord compression, bone radiation, and bone surgery as SREs. Why did authors exclude hypercalcemia from SREs? 

Thank you for your comment. Unfortunately, in health insurance claim data, it was impossible to retrieve the value of hypercalcemia. In terms of definition of SRE, some papers included hypercalcemia, but some papers did not include hypercalcemia depending on authors.

And what was the procedure cord for bone surgery? 

We added procedure codes for bone surgery in study population.

How did authors handle overlapping data from the different cords? 

We count 1st SRE event as index event for individual patient.

I don’t understand this sentence “In addition, patients diagnosed in 2002 were excluded for washout and were followed up for at least 2 years to assess for bone metastases and SREs”.

Our database is from 2002. In the first year, usually we use wash out period for cancer diagnosis because previous prevalent cases may confound incidence. We revised the sentence;

“We used a 1-year washout period to exclude cancer patients who had been diagnosed in the past.”

4. In result, authors commented that bone mets increased with age and there was no difference in region or household. Statistical p value should be added. If authors did not use statistical analysis, the content should be corrected.

Thank you for your critical comments. As your advice, we added statistical p value in table 2 and corrected my manuscript.

“Bone metastases showed an increasing trend in rural areas compared to that in urban areas. Household income had no impact on the occurrence of bone metastasis.”

5. Discussion is ambiguous and extensive. This cloud the issue. Please clarify authors’ suggestion and present specified contents.

Thank you. We revised the discussion part.

6. The manuscript is not easy to read and the English used really should be checked for grammar and spelling.

We requested English proofreading once again and submit it with completion. Thank you.

---

## [Decision Letter · Decision Letter 1]

5 Jun 2020

Bone metastasis and skeletal-related events in patients with solid cancer: a Korean nationwide health insurance database study

PONE-D-20-03800R1

Dear Dr. Hong,

We’re pleased to inform you that your manuscript has been judged scientifically suitable for publication and will be formally accepted for publication once it meets all outstanding technical requirements.

Kind regards,

Jason Chia-Hsun Hsieh, M.D. Ph.D

Academic Editor

PLOS ONE

Additional Editor Comments (optional):

All the questions were answered adequately.

Reviewers' comments:

Reviewer's Responses to Questions

**Comments to the Author**

1. If the authors have adequately addressed your comments raised in a previous round of review and you feel that this manuscript is now acceptable for publication, you may indicate that here to bypass the “Comments to the Author” section, enter your conflict of interest statement in the “Confidential to Editor” section, and submit your "Accept" recommendation.

Reviewer #1: All comments have been addressed

Reviewer #2: All comments have been addressed

Reviewer #3: All comments have been addressed

Reviewer #4: All comments have been addressed

2. Is the manuscript technically sound, and do the data support the conclusions?

Reviewer #1: Yes

Reviewer #2: Yes

Reviewer #3: Yes

Reviewer #4: Yes

3. Has the statistical analysis been performed appropriately and rigorously? 

Reviewer #1: N/A

Reviewer #2: Yes

Reviewer #3: Yes

Reviewer #4: Yes

4. Have the authors made all data underlying the findings in their manuscript fully available?

Reviewer #1: Yes

Reviewer #2: Yes

Reviewer #3: Yes

Reviewer #4: Yes

5. Is the manuscript presented in an intelligible fashion and written in standard English?

Reviewer #1: Yes

Reviewer #2: Yes

Reviewer #3: Yes

Reviewer #4: Yes

6. Review Comments to the Author

Reviewer #1: The authors have addressed questions posed by reviewers. I recommend acceptance of this revised version

Much improvement in English grammar

Reviewer #2: The authors have addressed all the issues raised sufficiently. This manuscript may be now acceptable for publication.

Reviewer #3: I reviewed the revised manuscript submitted by the authors.

The authors responded appropriately to what I pointed out.

Reviewer #4: (No Response)

7. PLOS authors have the option to publish the peer review history of their article (what does this mean?). If published, this will include your full peer review and any attached files.

Reviewer #1: No

Reviewer #2: No

Reviewer #3: No

Reviewer #4: No

---

## [Editor Report · Acceptance letter]

10 Jun 2020

PONE-D-20-03800R1 

Bone metastasis and skeletal-related events in patients with solid cancer: a Korean nationwide health insurance database study 

Dear Dr. Hong:

I'm pleased to inform you that your manuscript has been deemed suitable for publication in PLOS ONE. Congratulations! Your manuscript is now with our production department. 

Kind regards, 

on behalf of

Dr. Jason Chia-Hsun Hsieh 

Academic Editor

PLOS ONE